# Prevalence of Human Papillomavirus in Different Mucous Membranes in HIV Concordant Couples in Rwanda

**DOI:** 10.3390/v15041005

**Published:** 2023-04-19

**Authors:** Schifra Uwamungu, Bethelehem Nigussie, Claude Mambo Muvunyi, Bengt Hasséus, Maria Andersson, Daniel Giglio

**Affiliations:** 1Department of Pharmacology, Sahlgrenska Academy at the University of Gothenburg, SE-40530 Gothenburg, Sweden; schifra.uwamungu@gu.se; 2Department of Biomedical Laboratory Sciences, School of Health Sciences, College of Medicine and Health Sciences, University of Rwanda, Kigali P.O. Box 3286, Rwanda; 3Department of Pathology, Armauer Hansen Research Institute, Addis Ababa P.O. Box 1005, Ethiopia; bethelehemnigussie@gmail.com; 4Department of Clinical Biology, School of Medicine and Pharmacy, College of Medicine and Health Sciences, University of Rwanda, Kigali P.O. Box 3286, Rwanda; claude.muvunyi@rbc.gov.rw; 5Rwanda Biomedical Center, Kigali P.O. Box 7162, Rwanda; 6Department of Oral Medicine and Pathology, Institute of Odontology, Sahlgrenska Academy at the University of Gothenburg, P.O. Box 450, SE-40530 Gothenburg, Sweden; bengt.hasseus@odontologi.gu.se; 7Clinic of Oral Medicine, Public Dental Service, SE-40233 Gothenburg, Sweden; 8Department of Infectious Diseases, Institute of Biomedicine, Sahlgrenska Academy at the University of Gothenburg, SE-40530 Gothenburg, Sweden; maria.andersson.3@gu.se; 9Department of Clinical Microbiology, Sahlgrenska University Hospital, SE-41346 Gothenburg, Sweden; 10Department of Oncology, Institute of Clinical Sciences, Sahlgrenska Academy at the University of Gothenburg, SE-41345 Gothenburg, Sweden; 11Department of Oncology, Sahlgrenska University Hospital, SE-41345 Gothenburg, Sweden

**Keywords:** human papillomavirus, HIV, mucous membrane, cervical cancer, Rwanda

## Abstract

Background: The prevalence of human papillomavirus (HPV) infections in other anatomical sites besides the uterine cervix is unknown in East Africa. Here, we assessed the prevalence and concordance of HPVs in different anatomical sites in HIV concordant couples in Rwanda. Methods: Fifty HIV-positive concordant male-female couples at the HIV clinic at the University Teaching Hospital of Kigali in Rwanda were interviewed, swabbed from the oral cavity (OC), oropharynx (OP), anal canal (AC), vagina (V), uterine cervix (UC) and penis. A pap smear test and a self-collected vaginal swab (Vself) were taken. Twelve high-risk (HR)-HPVs were analyzed. Results: HR-HPVs occurred in 10%/12% in OC, 10%/0% in OP and 2%/24% in AC (*p* = 0.002) in men and women, respectively. HR-HPVs occurred in 24% of UC, 32% of Vself, 30% of V and 24% of P samples. Only 22.2% of all HR-HPV infections were shared by both partners (κ −0.34 ± 0.11; *p* = 0.004). The type-specific HR-HPV concordance was significant between male to female OC-OC (κ 0.56 ± 0.17), V-VSelf (κ 0.70 ± 0.10), UC-V (κ 0.54 ± 0.13), UC-Vself (κ 0.51 ± 0.13) and UC-female AC (κ 0.42 ± 0.15). Conclusions: HPV infections are prevalent in HIV-positive couples in Rwanda but concordance within couples is low. Vaginal self-sampling for HPV is representative of cervical HPV status.

## 1. Introduction

Chronic infection by high-risk human papillomaviruses (HR-HPVs) of the uterine cervix is the most important risk factor associated with the development of high-grade squamous intraepithelial lesions (HSIL) and cervical cancer where HPV16 and HPV18 stand for most cervical cancer cases [1]. Only a minority of HPV infections, however, become chronic, i.e., persisting for more than two years, since the immune system clears most HPV infections within a few months in immunocompetent individuals [2,3]. Co-infection with human immunodeficiency virus (HIV) may increase the risk of developing persistent HR-HPV infections and increases the risk of developing HPV-associated cancers [2,4,5]. HR-HPV infection is an important risk factor for the development of vaginal, penile, anal and head and neck cancer (HNC) [6]. In Western countries, there has been a dramatic increase in the incidence of HPV-associated oropharyngeal cancer squamous cell carcinoma (OPSCC), e.g., from before 2000 to 2009 the prevalence of HPV+ OPSCC increased from 40.5% to 72.2% [7]. The prevalence is particularly high in Sweden where it was reported that 85% of OPSCC in 2016 were HPV-positive [8]. The prevalence of HPV-positive OPSCC, however, varies significantly between countries. While developed countries report high prevalence figures for HPV+ OPSCC, HPV+ OPSCC seems to constitute less than 20% of all OPSCC in developing countries where prevalence figures also are uncertain for many countries [9].

In our previous study, we showed that 56% of all HR-HPV infections are chronic in Rwandan women living with HIV (WLWH) [2]. While cervical HPV infections and cervical cancer have been studied in most sub-Saharan countries, there is a lack of knowledge regarding the prevalence of HPV infections in other mucous membranes besides the uterine cervix. There is also a scarcity of knowledge on HPV co-infections between different anatomical sites among WLWH in East Africa. In this study, our primary aim was to assess the prevalence of HPV infections in different anatomical sites and to assess the concordance of HPV infections between HIV-positive concordant couples and co-infections between mucous membranes. Moreover, we examined the concordance between self-collected and physician-collected vaginal HPV screening. 

## 2. Materials and Methods

In this cross-sectional study, 50 HIV-positive concordant opposite-sex couples attending the HIV clinic at the University Teaching Hospital of Kigali (CHUK) in Kigali, Rwanda were recruited after informed consent. The couples lived together, and all participants were treated with antiretroviral therapy (ART). Inclusion criteria were literate individuals who were diagnosed with HIV and were 21 years old or older. Exclusion criteria were previous HPV vaccination, cervical lesions, women and men who have not had sexual intercourse or sexual activity with other individuals, and men and women for any reason considered not able to comply with the study protocol.

The participants were given a structured questionnaire including questions covering medical history, sexual and obstetric history, previous sexually transmitted infections (STIs), and questions covering risk factors for contraction of HPV infection. This was followed by taking a swab sample for HPV using the Aptima Multitest Swab (Hologic Inc., Marlborough, MA, USA) from the buccal mucosa, tonsillar area, and anal canal. In men, swab samples were taken from the glans penis/opening of the urethra. In women, swab samples for HPV were taken from the vagina by the woman herself and another vaginal sample was taken by the physician. This was followed by an HPV swab sample and a pap smear sample taken with a wood spatula from the uterine cervix taken by the physician. 

### 2.1. Real-Time PCR for Detection of HPVs

A real-time PCR assay targeting type-specific segments of the E6/E7 region for 12 high-risk (HPV16, 18, 31, 33, 35, 39, 45, 51, 52, 56, 58, and 59), two low-risk (HPV6 and 11) HPV types and beta-globin was used [10]. Nucleic acid from the 200 µL specimen was extracted by a MagNA Pure LC instrument (Roche Molecular, Mannheim, Germany) using the DNA Isolation I kit. The nucleic acids were eluted in 100 µL volume, and 5 µL of this were used for each real-time PCR reaction. Multiplex real-time PCR was performed in 8 parallel 20 µL-reactions containing oligonucleotides described in Appendix A, and Universal Mastermix (Applied Biosystems, Foster City, CA, USA). After 10 min of denaturation at 95 °C, the PCR was run for 45 cycles (15 s at 95 °C, 60 s at 58 °C) in a QuantStudio 6 384-well system (Applied Biosystems). The performance for each multiplex reagent mixture was evaluated using pUC57 plasmids with inserts of the targeted HPV sequences, synthesized by GenScript Corp. (Piscataway, NJ, USA). Combinations were only accepted if the cycle threshold (Ct) value in multiplex was no more than 1 cycle higher than detection carried out in separate reactions. The same plasmids were used in each run parallel with patient specimens to verify the performance of each target PCR. Only patient specimens with Ct values < 40 were considered positive. 

### 2.2. Cytology

Cytology of pap smears was classified according to the 2014 Bethesda System and the following cytological diagnosis: Negative for squamous intraepithelial lesion or malignancy (NILM), atypical squamous cells of undetermined significance (ASCUS), low-grade squamous intraepithelial lesion (LSIL), atypical squamous cells cannot exclude HSIL (ASC-H), atypical glandular cells (AGC), endocervical adenocarcinoma in situ (AIS), high-grade squamous intraepithelial lesion (HSIL) and squamous cell carcinoma (SCC).

### 2.3. Statistics

The SPSS version 24 (IBM) was used to assess the significance of differences detected in different main outcomes between and within groups. The chi-square test was used to compare categories. To compare the concordance of HR-HPV infections between different mucous membranes and between couples, Cohen’s kappa statistics were used. The Cohen’s kappa coefficient (κ) was estimated where κ < 0 = poor, 0–0.20 = slight, 0.21–0.40 = fair, 0.41–0.60 = moderate, 0.61–0.80 = substantial and 0.81–1.00 = almost perfect agreement. Bivariate logistic regression analysis was performed, and crude odds ratios (COR) and adjusted odds ratios (AOR) were calculated for explanatory variables correlating with being HR-HPV positive in any mucous membrane. Statistical significance was set to *p* < 0.05.

## 3. Results

### 3.1. Demographic Characteristics

Demographic data are displayed in Table 1. The median age of the 100 HIV-positive participants was 51.5 years (mean age 53.1 ± 9.1 years). Only 22% of men and women had passed secondary school. More women than men were unemployed (40% vs. 16%, respectively). The proportion of men who had or currently smoked and drank alcohol was significantly higher than that of women (*p* < 0.0001 and *p* = 0.017, respectively). Male participants reported a higher number of previous sex partners and orogenital sex partners than their female partners (*p* = 0.015 and *p* = 0.009, respectively). No participant reported same-sex sexual activity. Previous STIs were common among the participants, i.e., gonorrhea infection had occurred in 44% and 34%, syphilis in 18% and 24% and chlamydia in 12% and 12% of men and women, respectively. The proportion of men with CD4 counts lower than 200 cells/mL was significantly higher compared with women (*p* = 0.01), however, no difference was observed in HIV viral load between men and women. Ninety percent of men and women were in the agreement with the kind of contraceptive method used with their partner or in the agreement with not using a contraceptive method. Among 30 men reporting using contraceptive methods, 17% reported a different contraceptive method than their female partner or reported using a contraceptive method while their partner reported not using a contraceptive method.

### 3.2. Prevalence of HPV in Different Mucous Membranes and Cytology

The prevalence of different HPV types in different mucous membranes in men and women is displayed in Table 2. All analysed HPV types were detected in the cohort where HPV16, HPV52 and HPV35 were the most prevalent HR-HPV infections constituting 30%, 10% and 8%, respectively, of all HR-HPV infections in all assessed mucous membranes. HPV16 constituted 77%, 21%, 19%, 46%, 28–35% and 11% of all oral, oropharyngeal, anal, cervical, vaginal and penile HR-HPV infections, respectively.

The prevalence of LR- and HR-HPV infection-positivity in different mucous membranes in men and women are displayed in Table 3 and in Appendix A. Of the studied 100 participants, 45% were HR-HPV-positive in one or more mucous membranes. HR-HPV infections occurred in 10% of the oral cavity and oropharynx in men compared with 12% and 0% in women, respectively (*p* = 1.00 and *p* = 0.06, respectively). Oral and oropharyngeal LR-HPV infections were not detected in any of the study participants. Anal HR-HPV infections were significantly more common in women than men (24% vs. 2%, respectively, *p* = 0.002). In contrast, anal LR-HPV infections were rare in women and occurred to a similar degree in men (2% and 2%, respectively). Cervical HR-HPV and LR-HPV infections occurred in 24% and 6% of women, respectively. Vaginal HR-HPV and LR-HPV infections were detected with self-sampling in 32% and 4% of women, respectively, and with physician-taken sampling in 30% and 4% of women, respectively. A similar degree of genital HPV-positivity occurred also in men, i.e., penile HR-HPV and LR-HPV infections occurred in 24% and 4%, respectively. Co-infections of more than one HR-HPV type occurred in 1/5 male oral sample, 0/6 female oral sample, 2/5 male oropharyngeal sample, 0/1 male anal sample, 0/12 female anal sample, 1/12 cervical sample, 2/15 physician-taken vaginal sample, 2/16 vaginal self-sample and 5/12 penile sample. Low-grade and high-grade squamous intraepithelial lesions (LSIL and HSIL) in pap smears occurred in 8.7% and 4.3% of female genital HR-HPV-positive cases and in 11.1% and 0% of female genital HR-HPV-negative cases (Appendix A).

### 3.3. Concordance of HR-HPV Infections

HR-HPV type-specific concordances between mucous membranes and between couples are displayed in Table 4. Male oral to oropharyngeal HPV concordance was 87% (κ 0.46 ± 0.17; *p* < 0.001). Male-to-female oral HPV concordance was 90% (κ 0.56 ± 0.17; *p* < 0.001). The concordance of type-specific HR-HPVs in vaginal samples taken by the physician and by the participant herself was 87% (κ 0.70 ± 0.10; *p* < 0.001). The type-specific HR-HPV concordance between cervical samples and vaginal samples taken by the physician and the participant herself was 81% (κ 0.54 ± 0.13; *p* < 0.001) and 80% (κ 0.51 ± 0.13; *p* < 0.001), respectively. The type-specific HR-HPV concordance between cervical and female anal samples was 79% (κ 0.42 ± 0.15; *p* = 0.002). The type-specific HR-HPV concordance between vaginal self-sample and physician-sample vs. female anal samples was 74% and 72% (κ 0.36 ± 0.14; *p* = 0.005 and κ 0.30 ± 0.14; *p* = 0.02, respectively). Only 22.2% of HR-HPVs infections were shared by the couples and concordance was 35% (−0.34 ± 0.11; *p* = 0.004; Appendix A). Of the six couple-concordant HR-HPV infections, five were HPV16 and one was HPV39. 

### 3.4. Risk Factors Contracting HR-HPV Infections

Having more than two previous sex partners was associated with the risk of contracting HR-HPV infections in any mucous membrane in men in univariate analysis (*p* = 0.03) but not in multivariate analysis (*p* = 0.67; Table 5). Having more than one abortion and more than two previous sex partners, being previously infected by gonorrhea and not knowing if being infected by chlamydia were factors associated with the risk of contracting HR-HPV infections in any mucous membrane in women (Table 5).

## 4. Discussion

In the present study, we show that the prevalence of HPV infections was high in men living with HIV (MLWH) and WLWH, but HR-HPV infections were discordant within couples. While couples displayed concordant oral to oral HR-HPV infections, oral to genital HR-HPV infections were discordant. A high concordance between vaginal self-collected, vaginal physician-collected and cervical HR-HPV samples was observed. Previous exposure to STIs, the number of sex partners and undergoing abortions were factors correlating with the risk of being infected by HR-HPV. 

Our study and previous studies have shown a high prevalence of HPV infections in Rwandan men and women, especially in MLWH and WLWH [2,5,11,12,13]. The prevalence of HR-HPVs in the vagina/uterine cervix of 24–32% of WLWH was similar to the degree of positivity we previously observed at the same HIV clinic but in another cohort of women [11]. In the present study, we found that HPV16 was the most prevalent type followed by HPV-52 and HPV35, which is in accordance with our previous study analysing HPV in the uterine cervix in WLWH and HIV-negative women in Rwanda [11]. Few previous studies have been conducted analysing the prevalence of HPVs in the oral cavity and oropharynx in Africa. We show that 10–12% of our studied cohort were positive for oral group 1 HR-HPVs (IARC), which is in accordance with a cross-sectional study in Nigerian women but significantly higher than the prevalence in HIV cohorts from South Africa [14,15,16,17]. Oral to oral concordance was also high in couples, and HPV16 dominated among HPVs present in the oral cavity. Similarly, few studies have assessed the prevalence of penile HPV infections in sub-Saharan Africa. In a cross-sectional study conducted on Tanzanian men, one in five men were positive for HPV, and HIV constituted a risk factor for penile HPV [18]. In another study from Tanzania, the prevalence of penile HR-HPVs in MLWH was 25.7%, which was higher than the 15.8% observed in HIV-negative men [19]. We found similar rates (24%) for HR-HPV positivity in our cohort of MLWH. While HR-HPVs were rarely detected in male anal samples, HR-HPVs were commonly detected in female anal samples but detected to a less degree than in vaginal and cervical samples. Type-specific HR-HPV concordance between the three sites was high. A high concordance between anal and cervical HR-HPVs has been demonstrated in several previous studies [20,21,22,23]. 

In agreement with previous studies in sub-Saharan Africa [24,25,26], we demonstrate a high concordance between HR-HPVs detected in cervical and vaginal samples independent of whether the sample was collected by the patient herself or by the physician. The concordance between self- and physician-collected vaginal HR-HPV samplings was also high. The degree of concordance is in accordance with previous studies conducted in WLWH in Ethiopia and in other low-income countries [27,28]. Studies from Uganda have shown that self-administrated HPV testing is suitable and feasible in low-resource settings [29]. 

Besides a positive concordance between male and female oral HR-HPV strains, we observed a high discordance of type-specific sharing of HR-HPV infections between men and women. Our findings contrast with a larger study conducted in South Africa where HIV-concordant male-female partners shared HR-HPV strains [30]. In our questionnaire we did not cover extramarital sexual activity, however, men reported having more sexual partners, reported different contraceptive methods than their female partners and had more often than women multiple genital HR-HPV infections. 

We confirm again the relationship between gonorrhea infection and not knowing if being previously infected with chlamydia with HR-HPV infections in women [11]. To have had multiple sexual partners was correlated in women with the degree of HR-HPV-positivity in accordance with previous studies [31,32]. Voluntary/spontaneous abortion was found to be correlated with HR-HPV positivity in line with a meta-analysis showing a correlation between HPV infections and the risk of spontaneous abortion [33]. Taken together, exposure to unprotected sex correlates to the acquisition of HPV, however, we could not find a correlation between the use of condoms and protection against HPV. 

Rwanda is at the top in Africa when it comes to the vaccination of schoolgirls, reaching 89% of the target population [34]. Still, the majority of Rwandan women are unvaccinated and have never been screened for cervical cancer [11]. Moreover, the vaccination of boys has not been implemented in Africa and was only recently implemented in some but not all European countries and in the USA [35,36]. To further reduce the number of HPV-related cancers in East Africa we suggest that the focus should also be set to vaccinating boys in the future. 

The likelihood of being cervical cancer screened is higher in WLWH than in HIV-negative women in East Africa [37]. To note, in our included cohort of WLWH in 2021, 58% had undergone a pap smear test, which is an increase compared to the 27% who had undergone a pap smear in our previously studied cohort at the HIV clinic in 2015 [11]. The current percentage of women living with HIV who had been screened is higher than the one estimated for Rwanda in previous reports [37]. It is currently unknown whether the increase in cervical cancer screening in Rwanda also includes other women than risk patients for cervical cancer. 

The strength of our study is that we were able to analyse the concordance of HPV infections in multiple mucous membranes and analyse concordance of HPVs among HIV-positive couples. As far as we know, no previous study in East Africa has studied HPV co-infections in mucous membranes among risk patients. The present study has, however, also limitations. First, the cohort assessed was small and this may explain why no variable was related to HPV-positivity in men. Secondly, in our questionnaire, we did not cover extramarital sexual behaviour so we can only present indirect evidence for extramarital sexual activity contributing to the high degree of HR-HPV-positivity in men. Thirdly, we had a high average age of the participants, which most likely led to a lower degree of HPV positivity than in a younger cohort of individuals.

## 5. Conclusions

HPV infection is prevalent in mucous membranes among HIV-positive concordant couples in Rwanda. While oral to oral transmission was common in HIV-positive couples, genital infections were discordant. Vaginal HPV self-screening has, moreover, a high sensitivity to detect cervical HPVs. Our findings highlight the need for additional studies on HPV transmission in other mucous membranes than the uterine cervix in sub-Saharan Africa. In addition, the high prevalence of HPV concordance among HIV-positive concordant couples suggests that to further lower the HPV burden of disease in Africa, the focus should also be set on the vaccination of boys. 

## Figures and Tables

**Table 1 viruses-15-01005-t001:** Characteristics of patients (*N* = 100).

	Sex, *n* (%)		
Characteristics	Male 50 (50)	Female 50 (50)	*p*-Value
**Age**			**0.008**
30–39	2 (4)	5 (10)	
40–49	7 (14)	20 (40)	
50–59	23 (46)	19 (38)	
60–69	15 (30)	5 (10)	
70–79	3 (6)	1 (2)	
**Level of Education**			0.663
None	3 (6)	5 (10)	
Incomplete primary school	4 (8)	7 (14)	
Complete Primary school	13 (26)	15 (30)	
Incomplete Secondary school	19 (38)	12 (24)	
Complete secondary school	8 (16)	9 (18)	
Incomplete University	0 (0)	0 (0)	
Complete University	2 (4)	2 (4)	
**Occupation**			**0.001**
Farming	3 (6)	8 (16)	
Civil servant	1 (2)	0 (0)	
Business	10 (20)	13 (26)	
Community health worker	1 (2)	3 (6)	
Driver	4 (8)	0 (0)	
Tourism	3 (6)	4 (8)	
Security guard	4 (8)	0 (0)	
Technical worker	7 (14)	0 (0)	
Pastor	1 (2)	1 (2)	
Private sector	1 (2)	0 (0)	
Retired	7 (14)	1 (2)	
Unemployed	8 (16)	20 (40)	
**CD4 (cells/mL)**			**0.01**
<200	10 (20)	4 (8)	
200–500	29 (58)	21 (42)	
>500	11 (22)	25 (50)	
**Viral load (Virus/mL)**			0.459
<400	47 (94)	48 (96)	
400–5000	3 (6)	2 (4)	
>5000	0 (0)	0 (0)	
**Serious diseases**			0.221
No	27 (54)	33 (66)	
Yes	23 (46)	17 (34)	
Specified			
Cancer	3 (6)	2 (4)	
Cardiovascular	6 (12)	4 (8)	
Rheumatic disease	2 (4)	1 (2)	
Diabetes	5 (10)	4 (8)	
Respiratory tract disease	2 (4)	3 (6)	
Hepatitis B	1 (2)	1 (2)	
Other diseases	4 (8)	2 (4)	
**Smoking**			**<0.0001**
No	25 (50)	45 (90)	
Yes	2 (4)	0 (0)	
Ex-smoker	23 (46)	5 (10)	
**Use of alcohol**			**0.010**
Never	29 (58)	40 (80)	
Once a month	2 (4)	3 (6)	
Once a week	7 (14)	6 (12)	
Several times a week	12 (24)	1 (2)	
**Live births ***			
0	-	2 (4)	
1–2	-	16 (32)	
3–4	-	17 (34)	
5–6	-	9 (18)	
More than 6	-	6 (12)	
**Abortions ***			
0	-	27 (54)	
1	-	10 (20)	
2	-	9 (18)	
More than 2	-	4 (8)	
**First sexual intercourse**			0.425
10–14	2 (4)	0 (0)	
15–19	13 (26)	18 (36)	
20–24	23 (46)	23 (46)	
25–29	9 (18)	8 (16)	
30–35	3 (6)	1 (2)	
**Number of sex partners**			**0.012**
1–2	17 (34)	27 (54)	
3–4	10 (20)	15 (30)	
5–6	7 (14)	5 (10)	
7–8	3 (6)	0 (0)	
>8	13 (26)	3 (6)	
**Number of orogenital sex partners of the opposite sex**			**0.007**
0 partners	9 (18)	8 (16)	
1–2 partners	14 (28)	29 (58)	
3–4 partners	10 (20)	10 (20)	
5–6 partners	5 (10)	2 (4)	
7–8 partners	2 (4)	0 (0)	
>8 partners	10 (20)	1 (2)	
**Number of orogenital sex partners of the same sex**			1.00
0 partners	50 (100)	50 (100)	
1 partner and above	0 (0)	0 (0)	
**Number of anal sex partners of the opposite sex**			0.315
0 partners	49 (98)	50 (100)	
1–2 partners	1 (2)	0 (0)	
**Number of anal sex partners of the same sex**			1.00
0 partners	50 (100)	50 (100)	
1 partner and above	0 (0)	0 (0)	
**Previous gonorrhoea infection**			0.385
Yes	22 (44)	17 (34)	
No	28 (56)	32 (64)	
I do not know	0 (0)	1 (2)	
**Previous syphilis infection**			0.461
Yes	9 (18)	12 (24)	
No	41 (82)	38 (76)	
I do not know	0 (0)	0 (0)	
**Previous chlamydia infection**			0.697
Yes	6 (12)	6 (12)	
No	35 (70)	38 (76)	
I do not know	9 (18)	6 (12)	
**Previous other STI**			N/A
No	42 (84)	42 (84)	
Yes-specified			
Urinary tract infection	6 (12)	4 (8)	
Bacterial vaginosis	-	2 (4)	
Trichomonas vaginalis	-	2 (4)	
Not specified	2 (4)	-	
**Contraceptive methods**			N/A
Female sterilization	3 (6)	3 (6)	
Male sterilization	1 (2)	1 (2)	
Contraceptive implant	6 (12)	6 (12)	
Contraceptive injectable	3 (6)	4 (8)	
Contraceptive pills	1 (2)	2 (4)	
Male condom	12 (24)	8 (16)	
Rhythm method monitoring fertility	3 (6)	4 (8)	
Withdrawal of penis before ejaculation	1 (2)	0 (0)	
Other method	0 (0)	1 (2)	
None of them	20 (40)	21 (42)	
**Pap smear test**			N/A
Yes	-	29 (58)	
No	-	20 (40)	
I do not know	-	1 (2)	

Data are presented as number of individuals (%). *p*-values were calculated between males and females using the chi-square test. N/A = not applicable. * = Not-applicable to men.

**Table 2 viruses-15-01005-t002:** HR- and LR-HPV infections in different mucous membranes; n (%).

	Oral	Oropharynx	Anus	Cervix	Vagina -sc	Vagina -pc	Penis	Sum (% of All Infections)
HPV6			1 (1)	2 (4)	2 (4)	2 (4)	1 (2)	8 (7)
HPV11			1 (1)	1 (2)			1 (2)	3 (3)
HPV16	10 (10)	2 (2)	3 (3)	6 (12)	5 (10)	6 (12)	2 (4)	34 (30)
HPV18			1 (1)		1 (2)		4 (8)	6 (5)
HPV31	1 (1)	2 (2)					2 (4)	5 (4)
HPV33			1 (1)					1 (1)
HPV35			3 (3)	2 (4)	2 (4)	2 (4)		9 (8)
HPV39			1 (1)	1 (2)	2 (4)	1 (2)	1 (2)	6 (5)
HPV45			1 (1)				3 (6)	4 (4)
HPV51		1 (1)	1 (1)		1 (2)	2 (4)		5 (4)
HPV52	2 (1)	1 (1)		1 (2)	3 (6)	2 (4)	2 (4)	11 (10)
HPV56			1 (1)	1 (2)	2 (4)	2 (4)	1 (2)	7 (6)
HPV58			1 (1)	1 (2)	1 (2)	1 (2)	2 (4)	6 (5)
HPV59		2 (2)	1 (1)	1 (2)	1 (2)	1 (2)	2 (4)	8 (7)

Sc = self-collected; pc = physician-collected.

**Table 3 viruses-15-01005-t003:** Prevalence of HPVs in different mucous membranes.

Anatomical Site of HPV Infection	Yes/No	Males (*n* = 50)	Females (*n* = 50)	*p*-Value
Oral LR-HPV	Yes	0 (0)	0 (0)	1.00
	No	50 (100)	50 (100)	
Oral HR-HPV	Yes	5 (10)	6 (12)	1.00
	No	45 (90)	44 (88)	
Oropharynx LR-HPV	Yes	0 (0)	0 (0)	1.00
	No	50 (100)	50 (100)	
Oropharynx HR-HPV	Yes	5 (10)	0 (0)	0.06
	No	45 (90)	50 (100)	
Anus LR-HPV	Yes	1 (2)	1 (2)	1.00
	No	49 (98)	49 (98)	
Anus HR-HPV	Yes	1 (2)	12 (24)	0.002
	No	49 (98)	38 (76)	
Penis LR-HPV	Yes	2 (4)		
	No	48 (96)		
Penis HR-HPV	Yes	12 (24)		
	No	38 (76)		
Vagina LR-HPV (physician sampling)	Yes		2 (4)	
	No		48 (96)	
Vagina HR-HPV (physician sampling)	Yes		15 (30)	
	No		35 (70)	
Vagina LR-HPV (self-sampling)	Yes		2 (4)	
	No		48 (96)	
Vagina HR-HPV (self-sampling)	Yes		16 (32)	
	No		34 (68)	
Cervix LR-HPV	Yes		3 (6)	
	No		47 (94)	
Cervix HR-HPV	Yes		12 (24)	
	No		38 (76)	

Data are presented as number of individuals (%). *p*-values were calculated between males and females using the Fisher’s exact test. HPV = human papillomavirus; HR-HPV = high-risk human papillomavirus; LR-HPV = low-risk human papillomavirus.

**Table 4 viruses-15-01005-t004:** Concordances (κ) between mucous membranes and couples.

	M Oral	M Oropharynx	M Anus	Penis	F Oral	F Oropharynx	F Anus	Cervix	Vagina-sc	Vagina-pc
**M Oral**	1	**0.46** **±0.17 *****	−0.03 ±0.03	−0.03 ±0.11	**0.56** **±0.17 *****	No cases	−0.20 ±0.05	−0.08 ±0.11	−0.12 ±0.09	−0.13 ±0.09
	**M Oropharynx**	1	−0.03±0.03	0.04 ±0.12	0.02 ±0.14	No cases	−0.22 ±0.05	−0.21 ±0.05	**−0.25** **±0.07 ***	**−0.24** **±0.06 ***
		**M Anus**	1	0.07 ±0.07	−0.03 ±0.03	No cases	−0.04 ±0.04	0.11 ±0.10	−0.04 ±0.04	0.08 ±0.07
			**Penis**	1	−0.19 ±0.06	No cases	**−0.25** **±0.09 ***	−0.18 ±0.11	−0.36 ±0.09 **	**−0.24** **±0.10 ^#^**
				**F Oral**	1	No cases	0.07 ±0.14	−0.07 ±0.11	−0.004 ±0.11	−0.10 ±0.10
					**F Oropharynx**	No cases	No cases	No cases	No cases	No cases
						**F Anus**	1	**0.42** **±0.15 ****	**0.36** **±0.13 ****	**0.30** **±0.14 ***
							**Cervix**	1	**0.54** **±0.13 *****	**0.51** **±0.13 *****
								**Vagina-sc**	1	**0.70** **±0.10 *****
									**Vagina-pc**	1

M = male; F = female; SC = self-collected; PC = physician-collected; Statistical significance is indicated with * for *p* < 0.05, ** for *p* < 0.01, *** for *p* < 0.001 and **^#^**
*p* = 0.054.

**Table 5 viruses-15-01005-t005:** Risk of HR-HPV infections in any mucous membrane in men and women.

	Men				Women			
	COR (95% CI)	Sign.	AOR (95% CI)	Sign.	COR (95% CI)	Sign.	AOR (95% CI)	Sign.
**Age**								
≤51	Reference		Reference		Reference		Reference	
>51	0.52 (0.15–1.81)	0.31	0.38 (0.05-3.13)	0.37	2.12 (0.60–7.48)	0.25	9.80 (0.21–461.39)	0.25
**Level of Education**								
Less than complete secondary school	Reference		Reference		Reference		Reference	
Complete secondary school and beyond	1.67 (0.43–6.50)	0.46	2.18 (0.22–21.68)	0.51	1.03 (0.27–3.94)	0.97	3.01 (0.17–54.01)	0.45
**CD4 (cells/mL)**		0.65		0.20		0.31		0.09
>500	Reference		Reference		Reference		Reference	
200–500	1.14 (0.17–7.60)	0.89	0.51 (0.01–21.17)	0.72	1.27 (0.15–10.53)	0.82	0.02 (0–2.33)	0.08
<200	1.88 (0.41–8.60)	0.41	5.09 (0.34–76.07)	0.24	2.55 (0.76–8.48)	0.13	70.80 (0.45–11,134)	0.10
**Viral load (Virus/mL)**								
<400	Reference		Reference		Reference		Reference	
400–5000	3,446,346,331 (0-)	1.00	21,944,977,929, (0-)	1.00	0.85 (0.05–14.33)	0.91	18.30 (0–8,605,648)	0.66
**Smoking**								
Never	Reference		Reference		Reference		Reference	
Ever	1.00 (0.32–3.17)	1.00	1.44 (0.18–11.68)	0.73	1.31 (0.20–8.62)	0.78	1.89 (0.01–639.98)	0.83
**Use of alcohol**								
Never/seldom	Reference		Reference		Reference		Reference	
Once a month and more	2.39 (0.73–7.78)	0.15	4.98 (0.63–39.50)	0.13	0.49 (0.12–2.02)	0.33	2.16 (0.10–46.36)	0.62
**Live births**						0.36		0.22
0–2					Reference		Reference	
3–4					0.56 (0.15–2.14)	0.40	0.17 (0.06–4.92)	0.30
>4					1.60 (0.39–6.62)	0.52	6.46 (0.11–386.53)	0.37
**Abortions**								
0–1					Reference		Reference	
>1					7.22 (1.40–37.25)	**0.02**	276.39 (1.18–64,873)	**0.04**
**First sexual intercourse**								
<21	Reference		Reference		Reference		Reference	
≥21	0.71 (0.22–2.25)	0.56	0.82 (0.12–5.85)	0.84	1.21 (0.39–3.69)	0.74	0.43 (0.03–6.18)	0.53
**Number of sex partners**								
0–2	Reference		Reference		Reference		Reference	
>2	3.82 (1.13–12.95)	**0.03**	1.77 (0.13–23.67)	0.67	3.00 (0.54–16.60)	0.21	1478.09 (1.53–1,430,837)	**0.04**
**Number of orogenital sex partners**								
0–2	Reference		Reference		Reference		Reference	
>2	3.34 (0.96–11.62)	0.06	2.72 (0.22–33.95)	0.44	0.99 (0.28–3.52)	0.99	0.19 (0.01–5.93)	0.34
**Previous gonorrhoea infection**						0.21		0.10
No	Reference		Reference		Reference		Reference	
Yes	2.08 (0.64–6.73)	0.22	1.63 (0.19–14.19)	0.66	3.09 (0.88–10.83)	0.08	1042.63 (1.88–577,026)	**0.03**
I do not know					2,077,039,084 (0-)	1.00	89,632,519 (0-)	1.00
**Previous syphilis infection**						0.21		
No	Reference		Reference		Reference		Reference	
Yes	1.54 (0.36–6.67)	0.56	0.77 (0.03–18.51)	0.87	2.00 (0.51–7.78)	0.32	0.01 (0.00–2.42)	0.10
**Previous chlamydia infection**		0.25		0.71		0.25		0.10
No	Reference		Reference		Reference		Reference	
Yes	2.50 (0.43–14.54)	0.31	0.28 (0.00–18.95)	0.55	2.22 (0.36–13.62)	0.39	0.06 (0–20.71)	0.35
I do not know	3.13 (0.69–14.08)	0.14	1.54 (0.08–30.84)	0.78	5.56 (0.59–52.16)	0.13	23,199 (2.30–233,872,654)	**0.03**
**Contraceptive methods**								
Other method than condom/none	Reference		Reference		Reference		Reference	
Male condom	0.32 (0.06–1.68)	0.18	0.26 (0.02–3.57)	0.31	0.45 (0.09–2.13)	0.31	1.55 (0.03–72.71)	0.82
**Pap smear test**						0.84		0.92
No					Reference		Reference	
Yes					0.71 (0.22–2.22)	0.55	0.57 (0.04–8.97)	0.69
I do not know					0 (0-)	1.00	0 (0-)	1.00
**Partner HR-HPV positive**								
No	Reference		Reference		Reference		Reference	
Yes	1.57 (0.49–5.08)	0.45	0.82 (0.12–5.55)	0.84	1.57 (0.49–5.08)	0.45	17.93 (0.24–1320)	0.19

COR (crude odds ratio) and AOR (adjusted odds ratio) with 95% CI (confidence intervals) are indicated.

## Data Availability

Additional data can be obtained after written request to the Institute of Clinical Sciences, Sahlgrenska Academy at the University of Gothenburg, Medicinaregatan 3A SE-413 90 Göteborg, Sweden. Email: klinvet@gu.se.

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
