# Peer review of "Prevalence of Human Papillomavirus in Different Mucous Membranes in HIV Concordant Couples in Rwanda"

_viruses, 2023, doi:10.3390/v15041005_

Round 1

Reviewer 1 Report

In a well-written review, the authors describe the prevalence of high-risk HPV in mucosal samples from HIV positive concordant heterosexual couples from Rwanda. There was a low concordance in HPV types between couples, except for the oral cavity. Self-collected vaginal samples showed a good concordance with physician collected vaginal or cervical samples. Previous chlamydia or gonorrhoea infection and a higher number of sexual partners were positively associated with having high risk HPV infections.

The manuscript adds valuable novel data, the methods and results are clear; it is written in accessible language and the figures and tables facilitate understanding of the data.

The discussion cites relevant literature, and the strengths and limitations are recognized.

I have a few minor suggested edits.

1.      Abstract: Please write uterine cervix in full before abbreviating as “UC”

2.      Line 49: replace “infection” with “infections”

3.      Page 2 line 53-57: Please rephrase as Sweden is included amongst western countries. Perhaps state that ..."the prevalence is particularly high in Sweden…  "

4.      Table 3: Please include the “Male” heading

Author Response

Dear Reviewer 1,
Thank you for the valuable suggestions and comments. We have now changed the typos and errors in the manuscript as well as changed the sentence regarding prevalence of HPV+ OPSS to "The prevalence is particularly high in Sweden where it was reported that 85% of OPSCC in 2016 were HPV-positive".

Reviewer 2 Report

The study is useful because we need more information on HPV prevalence and transmission between couples - especially in Africa.

1.  "UC" is probably uterine cervix. Please define.

2. Some of the tables are excessive. Please shorten or eliminate and just refer to the data in the text.

3.  In 3.3, "HRP" may be a typo. 

4. The cytology seems to be quickly passed over - especially in men.

Author Response

Dear Reviewer 2,

Thank you for the valuable comments and suggestions. We have now changed the typos and errors in the manuscript.

  1. We have now changed "UC" to uterine cervix (UC) in the abstract
  2. We agree on that some of the tables are excessive. However, when we discussed what to include in the manuscript or in the supplemental materials we ended up with these five tables. We think it´s important that these five table are displayed in their full contents in the manuscript to easier navigate between the different covariates. 
  3. We changed the typo to "HR-HPV"
  4. The focus on the study was on HPV but we performed also pap smear on women. Cytology was only performed on pap smears. We have clarified this in Methodology as well as in Methods.